# Social Bots’ Involvement in the COVID-19 Vaccine Discussions on Twitter

**DOI:** 10.3390/ijerph19031651

**Published:** 2022-01-31

**Authors:** Menghan Zhang, Xue Qi, Ze Chen, Jun Liu

**Affiliations:** 1Centre for Chinese Urbanization Studies of Soochow University & Collaborative Innovation Center for New Urbanization and Social Governance of Universities, Suzhou 215006, China; mhzhang@suda.edu.cn; 2School of Communication, Soochow University, Suzhou 215123, China; 20205442018@stu.suda.edu.cn (X.Q.); 20204242011@stu.suda.edu.cn (Z.C.); 3Department of Communication, University of Copenhagen, DK-2300 Copenhagen, Denmark

**Keywords:** social bots, COVID-19 vaccine, social media analytics, twitter

## Abstract

During the COVID-19 pandemic, social media served as an important channel for the public to obtain health information and disseminate opinions when offline communication was severely hindered. Yet the emergence of social bots influencing social media conversations about public health threats will require researchers and practitioners to develop new communication strategies considering their influence. So far, little is known as to what extent social bots have been involved in COVID-19 vaccine-related discussions and debates on social media. This work selected a period of nearly 9 months after the approval of the first COVID-19 vaccines to detect social bots and performed high-frequency word analysis for both social bot-generated and human-generated tweets, thus working out the extent to which social bots participated in the discussion on the COVID-19 vaccine on Twitter and their participation features. Then, a textual analysis was performed on the content of tweets. The findings revealed that 8.87% of the users were social bots, with 11% of tweets in the corpus. Besides, social bots remained active over three periods. High-frequency words in the discussions of social bots and human users on vaccine topics were similar within the three peaks of discourse.

## 1. Introduction

As a platform promoting public engagement and discussion, social media is an important tool for the public to receive information, make judgments, and form cognition [1]. Intensive global efforts towards physical distancing and isolation to curb the spread of COVID-19 may have intensified the use of social media as individuals try to remain connected while in quarantine [2]. Social media has hence become an important channel for the public to understand and discuss vaccine-related information, acting as a channel tool to reflect and shape public perception [3].

Since the first COVID-19 vaccines were developed, social media has become one of the prominent spheres for the discussion and debate on vaccines, and information on social media will influence public attitudes towards vaccination [4]. Social media has played a central role in disseminating scientific information and online discussion during the current pandemic [5,6]. Nevertheless, increasing evidence has shown that the online content, discourse, and sentiment on social media may not be limited to genuine human-run accounts. In the earlier studies on vaccines, social bots—automated accounts controlled and manipulated by computer algorithms [7]—have been accused of participating in vaccine discussions on social media [8,9], including public sentiment manipulation [10,11]. “Weaponized” social bots can even pretend to be humans, spread false information on a large scale, and harm public health [8]. The normalization of social bots in social media may affect public perceptions of vaccine efficacy and shape public health knowledge [8], leading to a complicated opinion climate within social media and affecting public vaccination [12,13,14]. 

So far, scholars have found that social bots have become even more active in the era of COVID-19 [15,16]. To what extent social bots are participating in the discussion of COVID-19 vaccines online? Furthermore, the characteristics of their involvement have not been empirically investigated. The work detected social bots participating in the online discussion of the COVID-19 vaccine on Twitter and used textual analysis combined with a Python-based program to specifically analyze social bot-generated and human-generated tweets, thereby finding out the engagement characteristics of social bots. 

## 2. Literature Review

### 2.1. Social Media and Public Health Discourse

Social media is defined as web-based and mobile-based Internet applications allowing the creation, access, and exchange of ubiquitously accessible, user-generated content [17]. Information interaction on social media has the characteristics of real-time and anonymity of information dissemination [18], as well as the potential to reach a large audience and disseminate information rapidly [19,20]. These characteristics of social media make it an essential driving force for acquiring and disseminating information in different fields, e.g., in crisis management [21], politics [22], and dissemination of health information [5]. Analyzing data from social media offers new opportunities to analyze several aspects of health communication [12]. Kata found that at least 80% of Internet users seek health information online, and 16% of these seek online information about vaccination [23]. With the difficulty in communicating offline during the pandemic, more and more people are likely to use social media for information about vaccines and to express their views on social media platforms. Against this background, social media can provide an additional informal source of data that can be used to identify health information not reported to medical officials or health departments and to reveal viewpoints on related topics [18].

### 2.2. Vaccine Content on Social Media

With the spread of the epidemic, people are increasingly relying on different social media platforms to receive information and express opinions [10,24]. Starting in mid-2020, when the COVID-19 vaccine had not yet been developed, false information about the vaccine has been spread on social media platforms [25]. The U.S. Food and Drug Administration (2020) issued the Pfizer-BioNTech COVID-19 vaccine emergency-use authorization on December 11, 2020, implementing the COVID-19 vaccine as an essential measure to combat the COVID-19 epidemic. Discussions on the vaccine controversy have subsequently become one of the foci of online discussion [26]. 

Social media emerged as one of the main spheres for the debate and discussion of information related to COVID-19 and its vaccines [2]. Hussain et al. extracted over 300,000 social media posts related to COVID-19 vaccines, including 23,571 Facebook posts from the United Kingdom and 144,864 from the United States, along with 40,268 tweets from the United Kingdom and 98,385 from the United States from March 1 to November 22, 2020. Their study found that the public is optimistic about vaccine development, effectiveness, and trials, as well as concerns about vaccine safety and economic feasibility [27]. However, some studies show that the spread of conspiracy theories, false information, and various rumors about the COVID-19 vaccine on social media has caused an “infodemic” [28].

During the COVID19 pandemic, social media has become the main channel for disseminating official, authoritative information [29], rumor-defying information, and serious news [30]. Many government officials worldwide used Twitter as their primary communication channel to regularly share COVID-19-related policy updates and news [31]. 

### 2.3. Rise of Social Bots

In recent years, social bots have become increasingly prevalent on social media platforms. Social bots, as mentioned, are a program “that automatically produces content and interacts with humans on social media” [7]. These “automated social agents” [32] can mimic human users [33]. A study of social bots on Twitter shows that, of all English-speaking active users, 9 to 15% exhibit bot-like behaviors [15]. Shao et al. (2018) noted that a single social bot account can forward hundreds or even thousands of times in a short period, which enables social bots to be “super-spreaders” of information [34].

Scholars have detailed how social bots have been involved in politics [35,36], business [37], health [38], and other topics. In health-related topics, social bots have been accused of promoting polarized opinions, misleading information, and manipulating public sentiments [10]. For example, social bots can also influence the public’s attitudes towards the efficiency of cannabis in dealing with mental and physical problems [39]. In the COVID-19 pandemic, some pilots studies found that social bots spread and amplify false medical information and conspiracy theories [15].

### 2.4. Social Bots and Vaccine Discourse Online

Scholars have located social bot’s involvement in online vaccine discussions. Broniatowski et al. (2018) have identified “influencing bots” in a series of vaccine-related tweets designed to spread vaccine conflicts on Twitter. More specifically, “influencing bots” entail two types of inorganic users: accounts of “bots” generating automated content and those of “trolls” misrepresenting their identities and attempts to instigate conflicts purposefully [8]. Yuan et al. analyzed a retweet network related to the rubella vaccine on Twitter after the 2015 California Disneyland measles outbreak and confirmed the intervention of social bots. Their bot analysis discovered that 1.45% of the corpus users are identified as likely social bots, producing 4.59% of all tweets within the dataset. Moreover, bots may deepen the trend towards highly clustering anti-vaccine movements [9].

Without denying the contribution of these studies, researchers have failed to address the questions of how to distinguish social bots from human users and which social bots circulate in vaccine-related discourse. Besides, it remains empirically understudied as to whether there are social bots within the online discussion of the COVID-19 vaccine, and what are the characteristics of social bots. This work detected the existence of social bots in COVID-19 vaccine-related discussion on Twitter and characterized social bots’ involvement. The research questions are as follows.

RQ1: How many social bots are participating in the COVID-19 vaccine discourse on Twitter?

RQ2: What are the characteristics of social bots’ involvement in the question of COVID-19 vaccines?

The following section will first address the methodological issue regarding social bots’ detection, text classification, and analysis.

## 3. Materials and Methods

### 3.1. Data Collection and Cleaning

The work took the discussion related to COVID-19 vaccines on Twitter as the content of the analysis. We selected one important time point as a timestamp for data collecting: the U.S. Food and Drug Administration (FDA) officially approved the COVID-19 vaccine produced by Pfizer as the beginning of data collection on 11 December 2020, which was an important beginning for COVID-19 vaccination. The end of data collection was on 31 August 2021.

When people discuss vaccines and vaccination on Twitter, they often use the names of pharmaceutical companies to represent their vaccines. From the 22 vaccines that have been approved for use worldwide, the four vaccines with the widest coverage were selected as keywords, including Pfizer, Moderna, Johnson & Johnson, and AstraZeneca [40]. We observed the hashtags of four pharmaceutical companies and their vaccines on Twitter. Therefore, a list of hashtags related to the COVID-19 vaccines was crafted, including #Pfizer vaccine, #Moderna vaccine, #Johnson & Johnson vaccine, and #AstraZeneca vaccine, which made our search results more comprehensive (# is the representation of hashtags on Twitter). For example, the relevant tweets of #Pfizer can also be retrieved when the #Pfizer vaccine is used for retrieval.

For selection of hashtags, when #vaccine was used as the retrieved hashtag, the tweets’ content included multiple vaccines. In other words, the public attaches the hashtag #vaccine to their tweets when discussing any vaccines, such as HPV vaccines or pet vaccines. The reason for not using #covid-vaccine and #covid vaccine is that the hashtag #covid-vaccine does not exist on Twitter. Searching with the hashtag #covid-vaccine shows tweets containing the keywords “covid” and “vaccine.” Using #covid vaccine also leads to search results with discussions related to #covid rather than covid-19 vaccines due to the space between “covid” and “vaccine,” deviating from our research direction. Using the specific hashtags of four pharmaceutical companies and their vaccines is thereby more focused on the research topic of this work, and we would like to see public discussions on more specific vaccines. In other words, more general hashtags are not used, which is our limitation. This issue will be studied in the follow-up research.

Then, the official API data interface service provided by Twitter was used to collect tweets, with all the tweets in these hashtags crawled. Twitter users’ privacy is respected without any personal information collected and displayed. A total of 314,342 vaccine-related tweets from 11 December 2020, to 31 August 2021, were obtained. Table 1 reports the complete list of search terms and the total number of tweets containing each hashtag. In the dataset we obtained, the number of tweets in English accounted for 96%, and that of non-English tweets accounted for 4%. Considering that English is the main language in the Twitter platform, we excluded non-English tweets from the dataset. We filtered out the repeated tweets, hyperlinks, and emoji symbols in the tweets. After data cleaning, 225,277 valid tweets were obtained.

### 3.2. Social Bot Detection

Social bots are identified by Botometer formerly (Indiana University, Bloomington, IN, USA), known as BotOrNot. The program was developed by the Institute of Network Science in Indiana University and uses a specific identification algorithm integrating user behaviors, relationships, and spatiotemporal data. It is currently the most authoritative software for identifying bot accounts [7,39,41]. As a machine-learning framework, Botometer extracts more than 1000 features of users’ behaviors through user profile data, language, social network structure, temporal activity, and sentiment. Then the program produces a score indicating the likelihood that a Twitter account is a bot. Scores closer to 1 represent a higher chance of being bots, while those closer to 0 are more likely to belong to humans [10]. Based on previous studies [7,39,41], the threshold was set to 0.5 to separate bots from humans in this work. A score above 0.5 was defined as a social bot.

### 3.3. Textual Analysis

The contents of tweets were analyzed to further explore what social bots and human users said in discussing the COVID-19 vaccines. Firstly, these 255,277 valid tweets were divided into two parts, i.e., posted by human users and by social bots, respectively. Besides, tweets posted by social bots and human users are classified by month.

Firstly, the high-frequency words used were summarized to understand the foci of the discussion between social bots and human users. Cipin, a professional word frequency analysis software based on Python, was applied, with NLTK Library used for word segmentation. NLTK is a leading platform for building Python programs to work with human language data. It provides easy-to-use interfaces to over 50 corpora and lexical resources such as WordNet, along with a suite of text processing libraries for classification, tokenization, stemming, tagging, parsing, and semantic reasoning, wrappers for industrial-strength NLP libraries, and an active discussion forum [42]. Loading American National Corpus, we used NLTK chunk nltk.tokenize for word segmentation. Firstly, nltk.sent_tokenize(text) was used to complete the segmentation between sentences, and nltk.word_tokenize(sentence) used to complete word segmentation.

The tweets of human and social bots were input to the software to discover word frequency statistics. We excluded these hashtags (#Pfizer vaccine, #Moderna vaccine, #Johnson & Johnson vaccine, and #AstraZeneca vaccine) used to retrieve tweets. After obtaining the statistical results, the top five high-frequency words were selected. Tweets containing these high-frequency words were retained and manual text analysis performed.

## 4. Findings and Discussion

### 4.1. Social Bots Participated in Online COVID-19 Vaccines Related Discourse

When detecting a large number of accounts, we used the API calling method provided by Botometer. The Python program was used to call the API, and the detection could obtain a value range of 0–1. The higher the score, the more likely the account was a social bot. Based on the judgment criteria of [7,10], the work adopted a conservative threshold to ensure the scientific nature of the research results, i.e., the threshold was set to 0.5 to distinguish between humans and social bots. If the score was higher than 0.5, the account is considered to be a bot.

We entered 255,277 valid tweets from 102,844 Twitter users into Botometer to obtain a score. Figure 1 shows the distribution of scores returned by Botometer. Botometer detects 9122 social bots and 93,722 human users among 102,844 Twitter users, accounting for 8.87% and 91.13% of total users, respectively. Social bots post 28,081 and humans post 227,196, accounting for 11% and 89% of the total tweets, respectively. This finding proves the first question that 8.87% of users who tweeted about the COVID-19 vaccine are social bots. Tweets posted by social bots and human users are analyzed to figure out what the social bots posted and how these actors influenced online COVID-19 vaccine discussions.

### 4.2. Three Booms in the Discourse of the COVID-19 Vaccine on Twitter

Regarding the number distribution of tweets, the change in the number of social bots’ tweets is consistent with that of humans. Meanwhile, Figure 2 shows that the number of tweets by human users increases by more than 25% in December–January 2020, March–April 2021, and June–July 2021. In other words, they have the highest discussion heat at these three stages. So what do social bots post when human users are arguing about a COVID-19 vaccine on social media? These three time periods are divided for investigation: December 2020 to January 2021 as the first phase of discourse (Stage 1), March to April 2021 as the second phase of discourse (Stage 2), and June to July 2021 as the third phase of discourse (Stage 3). Next, what humans and social bots discuss in these three periods is specifically analyzed.

### 4.3. Discourse Foci of Social Bots and Human Users

The tweets released by social bots and humans in these three periods were analyzed to further explore the three periods mentioned above. Table 2 presents high-frequency words of social bots and human users in the three periods. According to the statistics of high-frequency words, the foci of the three concentrated discussions are different. 

Stage 1: Discourse on the Effect of Vaccines (from December 2020 to January 2021).

Word-frequency statistics were used to obtain the high-frequency words in social bots and human tweets. Four of the five most frequent words in social bots and human tweets coincided: Dose, Effective, Trial, and Against. Their discussions focused on whether the COVID-19 vaccines were effective. As shown in Table 3, tweets posted by humans during the Stage1, human users expressed more concerns about vaccine risks in their tweets, such as concerns about serious side effects of vaccine experiments (example 1), doubts about vaccine effectiveness (example 3), and concerns about the adequacy of vaccine quantity (example 5). On the contrary, social bots conveyed positive information that a variety of COVID-19 vaccines could be gradually approved, and the number of vaccines could be guaranteed (examples 2 and 4). They explained that the vaccines were highly effective after rigorous scientific trials and vaccination could resist COVID-19 (example 6). Besides, social bots could connect efficient vaccines with various positive words, such as Freedom (example 7), Protective (example 8), Hope (example 9), and others, by emphasizing the immunization rate after vaccination. While emphasizing the effectiveness of vaccines, they could describe life after vaccination very positively.

Stage 2: Disputes on “Blood Clots” (from March to April 2021).

Unlike the previous discussion, from March to April 2021, the focus of the discussion changed to “blood clots” caused by the COVID-19 vaccination. In this period, the words most used by humans are News, Blood Clots, Pause, CDC, and Health, and the words most used by social bots are Blood Clots, Pause, CDC, News, and Biden. Table 4 shows the typical tweets posted by social bots and human users at the Stage 2.

In example 10, some users believed that other vaccines might also cause “blood clots” or other side effects, but this had not yet been confirmed. At stage 2, social bots introduced the vaccine’s efficacy as in stage 1 and joined the discussion of “blood clots”. In response to the public’s concern that “infection with COVID-19 did not kill, but people died after COVID-19 vaccination (examples 12 and 14)”, social bots used the number of people who get “blood clots” after vaccination as the support for the argument, adopting the discourse “advantages outweigh disadvantages”. For example, they emphasized that “Two in one- million women who have had the COVID-19 vaccine have had blood clots (example 11)”. Besides, they believed that the consequences of vaccine side effects were not worth mentioning compared with the severe consequences caused by the failure of the immune barrier (example 13).

Although both social bots and human users had actively participated in discussing “blood clots”, there were differences in their discourse. When using #CDC to indicate the information source of tweets, human users sent more CDC notifications about the risk of “blood clots” (example 16), while the social bots skillfully avoided this topic. They repeatedly emphasized that the COVID-19 vaccines had high protection and showed that CDC had unsealed the Johnson & Johnson vaccine under investigation, indicating that it was still safe (example 15). Interestingly, the word “Biden” also appears in a large number of social bot tweets. In example 17, tweets of social bots containing “Biden” described one topic: Biden had purchased 500 million Pfizer vaccines and would make a “historic” donation. These vaccines would be donated to 92 developing countries to alleviate the vaccine gap caused by worldwide, uneven vaccine distribution.

Stage 3: Debate on Delta (from June to July 2021).

In July 2021, the discussion on COVID-19 vaccines climbed to a peak again. At this stage, the high-frequency words from social bots are Delta, Against, FDA, Warning, and Effective, while the high-frequency words from human users are Delta, Against, FDA, Health, and Warning. Although the high-frequency words of social bots are highly consistent with human users, there are differences in their description of whether COVID-19 vaccines can effectively resist the Delta variant. Table 5 shows the typical tweets posted by social bots and human users at the Stage 3.

The public questioned the effectiveness of existing vaccines in dealing with the Delta variant (examples 18 and 20). Yet, social bots advocated that COVID-19 vaccines were effective against the Delta variant (example 19). They said that the immune response could last at least eight months after vaccination (example 21). If people wanted to get better protection under the spread of the Delta variant, they could inject reinforcing needles (example 22). Besides, social bots were more inclined to convey such a message to human users without being fully vaccinated. If the existing vaccine was only vaccinated with one shot or not fully vaccinated, the protection rate and practical utility were limited (example 23).

### 4.4. Three COVID-19 Vaccines Discussion Peaks Linked to Real-World Events

Vaccine discussions on social media are constantly evolving, with trends linked to real-world events [43]. Our research further confirms this view. The three critical discussion periods were closely related to actual events. The three critical discussion periods were closely related to actual events. We show the relationship with Figure 3. People mainly discussed whether the vaccine was effective and whether the vaccine cost money from December 2020 to January 2021, because the COVID-19 vaccines had just come out. On December 11 and 18, 2020, the FDA issued emergency-use authorizations for Pfizer and Moderna vaccines to prevent COVID-19. COVID-19 mRNA vaccines were officially released to the public as an effective tool to combat the epidemic. However, could vaccines resist COVID-19? How could vaccines be obtained? Were the vaccines free? The public did not know much about the COVID-19 vaccines. They alleviated their anxiety in this information environment full of uncertainty and crisis by communicating and forwarding news on Twitter.

On 16 April 2021, AstraZeneca COVID-19 vaccines were reported to have produced a rare “blood clots” event. Meanwhile, the government proposed to suspend the company’s COVID-19 vaccine production because six women vaccinated with Johnson & Johnson had rare “blood clots” and one woman died. The “blood clots” event aroused people’s concerns and fears about the COVID-19 vaccines, which were discussed fiercely on social media. 

In July 2021, the Delta virus was found in at least 98 countries and regions. According to the CNN June 29 report, the number of cases of COVID-19 infection in the United States was increasing. The reported cases almost covered the whole of the United States. The Delta virus has become the world’s leading strain of COVID-19, which has increased the number of new cases and deaths worldwide. At this time, the discussion on social media also turned to the discussion of the delta mutant strain.

## 5. Conclusions

In recent years, the rise of social bots has attracted the attention of scholars [7,41,44]. Researchers found that these automated agents have become even more active in the era of COVID-19 [10,17]. Social bots have been found in the discussion on the COVID-19 epidemic on Twitter [10,16]. The emergence of bots influencing social media conversations about public health threats will require researchers and practitioners to develop new communication strategies that take into account the influence of bots [45].

The work collected 314,342 tweets from 21 December 2020, to explore whether social bots participated in the discussion of the COVID-19 vaccines on Twitter. Our research was launched when the FDA approved the Pfizer vaccine, the first batch of COVID-19 vaccines in the world, on 30 August 2021. Nine months of observation and analysis were used to track the changes in this topic discussion. The number of tweets on vaccine topics by humans and social bots fluctuated. Data collection over a long period enables us to find these differences and changes between human users and social bots in tweet content and expression while observing discussion on COVID-19 vaccines. Then, Botometer was used to detect social bots. Our findings reveal and thereby enrich current discussions on social bots from the following three aspects.

(1) Our findings evidenced the participation of social bots in the discussion of COVID-19 vaccines on Twitter. We detected 28,081 (11%) tweets published by 9122 (8.87%) social bots in the dataset. (2) The activity level of social bots was consistent with that of human users. The number of tweets increased significantly from December 2020 to January 2021, March to April 2021, and June to July 2021. (3) Social bots and humans remaining active in these three periods were mainly relating their discussion to actual events. 

From December to January 2020, human users were still skeptical about the COVID-19 vaccines because they had just come out. However, social bots began to promote the effectiveness of the vaccines actively. From March to April 2021, the “blood clots”, caused by AstraZeneca, a side effect after vaccination, led to public panic about the COVID-19 vaccines. Although social bots began to talk about “blood clots” at this stage, their discourses were opposite to those of human users. By citing positive sources from CDC, social bots emphasize that “blood clots” were not a common problem. From June to July 2021, the Delta virus ravaged the world. Human users were worried about whether the existing COVID-19 vaccines could resist the Delta virus, but social bots thought that the vaccines were effective against the Delta virus. Generally, social bots are optimistic towards vaccines, although social bots have not always discussed the effectiveness and importance of vaccines.

To conclude, the work enriches the existing research on social bots and online vaccine discussions and helps to comprehensively understand social bots’ characteristics and current dynamics. With the spread of the COVID-19 epidemic, vaccination against COVID-19 has become an important means to control the epidemic. The normalization of social bots in social media may affect public perceptions of vaccine effectiveness and shape public health knowledge, complicating the climate of opinion on social media. Researchers and practitioners need to understand the engagement of social bots in social media to develop new communication strategies. We hope that, through our study, researchers and policymakers can understand how social bots are participating in COVID-19 vaccine discussions.

## 6. Limitations and Future Directions

The work detected the participation of social bots in the online discussion of the COVID-19 vaccine on Twitter. Textual analysis was used to specifically analyze bot-generated and human-generated tweets, thus finding out the engagement characteristics of social bots. However, while bot detection algorithms can automatically label suspicious social bots, the Twitter platform’s current mechanism does not reveal the manipulators behind social bots. Therefore, although characterizing the content posted by social bots and their participation in discussions and proposing several possible explanations, the work cannot infer the intentions of social bots based on their expressions. Previous researchers have suggested that social bots can also influence people’s emotions about vaccines [9,10]. Therefore, our subsequent research will focus on the emotional expression of social bots in the discussion of the COVID-19 vaccine and whether they affect the public’s sentiments towards the COVID-19 vaccine. 

## Figures and Tables

**Figure 1 ijerph-19-01651-f001:**
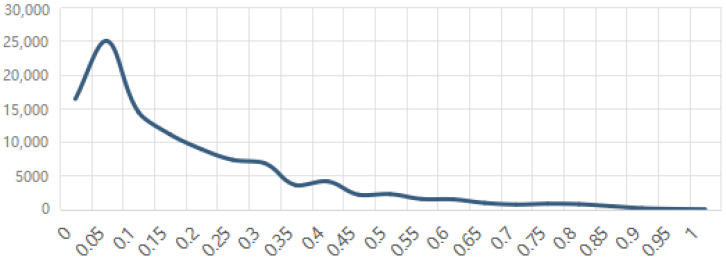
Score Distribution Returned by Botometer.

**Figure 2 ijerph-19-01651-f002:**
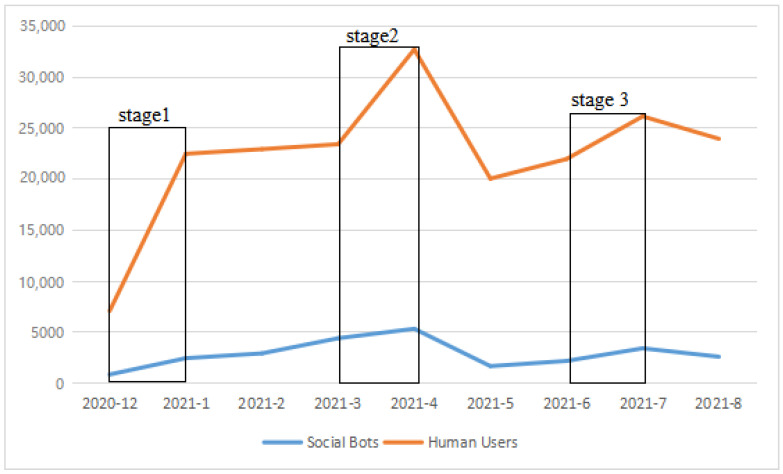
Number of monthly tweets of bots and Humans from December 2020 to August 2021.

**Figure 3 ijerph-19-01651-f003:**
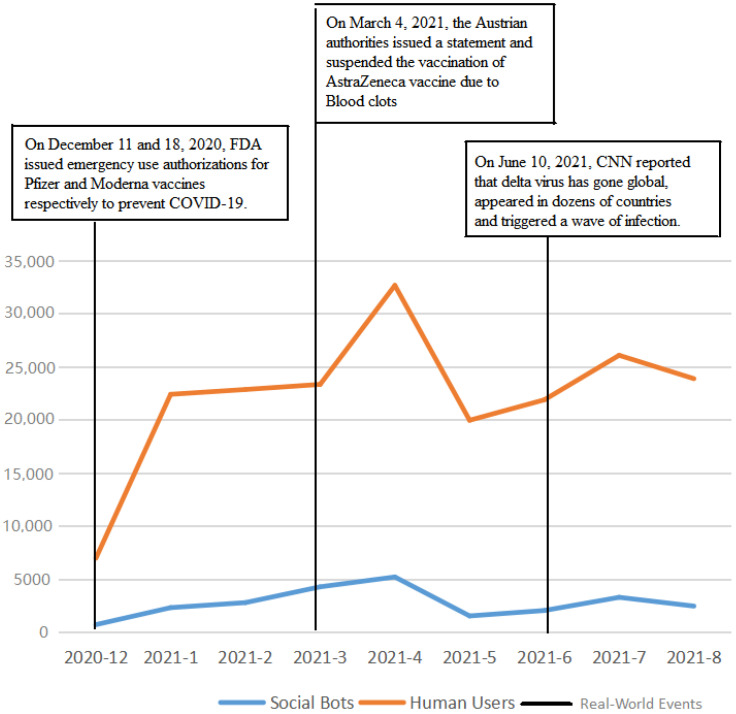
Trends of Tweets Published by Social Bots and Human Users Related to Real-World from December 2020 to August 2021.

**Table 1 ijerph-19-01651-t001:** List of Hashtags Related to the COVID-19 Vaccination.

Hashtags	All Tweets	Valid Tweets
#Pfizer vaccines	63,641	45,609
#Moderna vaccines	54,109	38,777
#Johnson & Johnson vaccines	139,823	100,205
#AstraZeneca vaccines	56,769	40,686
Summation	314,342	225,277

**Table 2 ijerph-19-01651-t002:** High-Frequency Words of Social Bots and Human Users in Three Periods.

2020.12–2021.1						
	Dose	Effective	Against	Trial	EU	News
Human users	10,398	8841	5985	6477	6353	-
Social bots	979	1282	714	516	-	821
**2021.3–2021.4**						
	News	Blood clot	Pause	CDC	Health	Biden
Human users	8275	7224	6922	3888	7422	-
Social bots	608	1726	1535	917	-	710
**2021.6–2021.7**						
	Delta	Against	FDA	Warning	Health	Effective
Human users	8021	7710	7213	4531	6049	-
Social bots	1957	1869	1590	1416	-	1018

**Table 3 ijerph-19-01651-t003:** Typical Tweets—Examples of Social Bots and Human Users at the Stage 1.

Stage	Humans	Bots
Stage 1	Example 1: #COVID19 vaccine side effects: What it is like in #Pfizer, #Moderna trials The FDA says Pfizer’s COVID vaccine is safe and effective. However, trial participants warn of intense…	Example 2: #AstraZeneca to produce 90 million #COVID19 vaccine shots in Japan East Asia News @Straits_Times AstraZeneca to produce 90 million COVID-19 vaccine shots in Japan, East Asia News -Announcement comes as concerns mount over whether the country will have enough to start its delayed inoculation drive. Read more at straitstimes.com
	Example 3: I heard that COVID 19 $AZN #AstraZeneca/#Oxford #COVID19 AstraZeneca vaccine is reportedly only 8% effective in populations older than 65…	Example 4: #Pfizer BioNTech #Astrazeneca#Pfizer”“One step closer to freedom. You can help stop the pandemic by getting a COVID-19 vaccine when it is available to you. Join the global effort! Together against the COVID-19!
	Example 5: #AstraZeneca offers eight million more doses of its #coronavirus vaccine to the #EU to try to defuse a row over supplies, but the bloc says that was too far short of what was originally promised, according to an EU official.	Example 6: Did you know all Phase 1 trials test several dose levels of vaccines? The dose levels carried forward to Phase 2; 3 in the #Pfizer and #Moderna trials are the best balance of safety and efficacy. Sign up for no cost COVID-19 updates: https://bit.ly/3rk9QP9 (accessed on 14 December 2021)
	-	Example 7: #PfizerBioNTech #astrazeneca#Pfizer”“One step closer to freedom. You can help stop the pandemic by getting a covid-19 vaccine when it is available to you.
	-	Example 8: #Moderna Vaccine Is Highly Protective Against COVID-19, the F.D.A. Finds #Moderna Says Vaccine Still Protects Against Virus Variants #Johnson; Johnson’s Vaccine Offers Strong Protection but Fuels Concern About Variants
	-	Example 9: Replying to @KulganofCrydee and @IainDale The first of many great Brexit success stories. Let’s hope the vaccines start to speed up across Europe so we can all have our freedoms returned. #Brexit #AstraZeneca #CovidVaccine

**Table 4 ijerph-19-01651-t004:** Typical Tweets—Examples of Social Bots and Human Users at Stage 2.

Stage	Humans	Bots
Stage 2	Example10: #Janssen vaccine also has ‘possible link’ to blood clots: https://onenewspage.com/video/20210420/13805259/ (accessed on 14 December 2021) Janssen-vaccine Has possible link to blood clots.htm (accessed on 14 December 2021).	Example 11: 2 in 1 million women who have had the COVID-19 vaccine have had blood clots, 1 in 5000 women taking the pill have blood clots, 1 in 6 people with COVID gets blood clots.
	Example 12: #Brazil#AstraZeneca A pregnant woman survived after contracting the virus, and died after receiving the#AstraZeneca vaccine. #bloodclot	Example 13: The Johnson& Johnson shot has been given to some 7.2 million Americans, with six cases of women developing brain clots along with low blood platelet counts, including one death, within two weeks of getting the one-dose vaccine.#MNow #Vaccine #USUS call on resuming J&J vaccine expected this week#Science #science How The Johnson & Johnson Vaccine Pause Might Slow Down The Global COVID-19 Response? How The Johnson & Johnson Vaccine Pause Might Slow Down The Global COVID-19 Response?
	Example14: Survived after contracting the COVID-19 virus, and died after receiving the #AstraZeneca vaccine.	Example15: #Breaking: #US ends pause on #Johnson; Johnson #Covid vaccine after concern about clots#Breaking: #US ends pause on #Johnson & Johnson #Covid vaccine after concerning about clots (http://Forbes.com) (accessed on 14 December 2021):#Fauci #Denies Johnson; Johnson Pause Will Contribute To Vaccine Hesitancy: “In the long run … people will realize that we take safety very seriously,” Fauci said… #TrendsSpy http://newsoneplace.com/14648812012/fauci-denies-johnson-pause-will-contribute-vaccine (accessed on 14 December 2021)…
	Example 16: CDC panel reviewing Johnson & Johnson vaccine pause Friday. The CDC is investigating 2 new reports of clotting tied to the Johnson & Johnson COVID-19 vaccine.-	Example17: US President Joe Biden is expected to announce a “historic” US donation of half a billion COVID-19 vaccine doses for 92 poorer countries, the White House said #News #Pfizer #Vaccine #Coronavirus #Worldh24.news

**Table 5 ijerph-19-01651-t005:** Typical Tweets—Examples of Social Bots and Human Users at Stage 3.

Stage	Humans	Bots
Stage 3	Example 18: A study from a team of New York University researchers found the one-shot Johnson & Johnson vaccine is far less effective at preventing coronavirus infections from the Delta variant and other mutated…	Example19:the Johnson & Johnson Coronavirus vaccine provides effective protection against the Delta variant, which is offering hope to many developing economies facing a summer surge of the highly contagious…
	Example 20: Israel study suggests #Pfizer/BioNTech #vaccine is less effective against #Delta #variant. Data suggests the vaccine is 64% effective at preventing infection caused by the Delta variant after two doses. Israel study suggests Pfizer/BioNTech vaccine is less effective against Delta variant.	Example 21: Johnson & Johnson asserts that its single-dose Janssen COVID-19 #vaccine is effective against the #Delta variant, which was first identified in India and is particularly contagious, with an immune response that can last at least eight months. (AFP)
	-	Example 22: #MOG Pfizer’s COVID-19 vaccine works very well against the Delta variant—but only after 2 doses. The study on 19,000 people in the UK found one Pfizer shot was about 36% effective against Delta, but it was highly effective after two shots. businessinsider.com
	-	Example 23: #Pfizer’s COVID-19 vaccine works very well against the Delta variant—but only after 2 doses. The study on 19,000 people in the UK found one Pfizer shot was about 36% effective against Delta, but it was highly effective after two shots.

## Data Availability

The data presented in this study are available on request from the first author. The data are not publicly available due to privacy protection.

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
