# Peer review of "Social Bots’ Involvement in the COVID-19 Vaccine Discussions on Twitter"

_ijerph, 2022, doi:10.3390/ijerph19031651_

Round 1

Reviewer 1 Report

Abstract and keywords:

Write a brief and summarized sentence or two about implications of the current study. It is good to see summary of important findings here in the abstract, however, it needs to be complemented by a reasonable conclusion, in a way that the reader can not only understand the rationale of the study but have some idea about what to expect in the following sections. Also, what specific method(s) are used in this research? That being explained, the abstract has rooms for improvement and needs to be enriched by the aforementioned elements.

Keywords look good, but could better represent the current study (e.g., Corona vaccine instead of vaccine), also consider adding some other relevant keywords such as social media analytics, Twitter, etc.)

Introduction:

Introduction is somehow shallow and requires more input on the following: 1) describing the background and introducing research problem: explain in a sentence or two, the significance/implications/impacts of social media conversations on COVID vaccination, elaborating more on your very first citation (Benis et al., 2021), also briefly explain the importance of social bots in such discussion and basically in what ways they would manipulate vaccine discussions (a condensed summary of section 2.3); 2)what are specific objectives of the current study? what is the salience of your research? What are some main methods used for investigation?

Literature review:

Literature review is written well, however, author(s) are highly encouraged to step back and assign a sub-section to social media, social media conversations, social media analytics, and other broad concepts in the first place, before starting with vaccine content and social media. A literature review on social media and social media analytics would be highly beneficial here and helps the lay reader to better understand/conceptualize the bigger picture. Another concern about the remainder of the lit review sections is that some major and highly related publications are missing and should be added accordingly (e.g., Charles-Smith et al., 2015).

Methodology

Briefly explain the previous research design that was followed in the current study. What exact locations (207 among 218) author(s) pointed out to, and basically what did they mean by “location,” is this geographical coordinates used to interact with Twitter API, or country/city/etc.?

How did the author(s) decide about vaccines with broadest coverage? Is this based on their own intuitions or are there specific scientific research that they can refer back to? And even more controversially, why more generally relevant hashtags such as #vaccine #covidvaccine are excluded from the search vocabulary?

Were English tweets were extracted only, and if so, what is the author(s) rationale for excluding non-English tweets?

How did the tweets were extracted in the first place? Was it through Twitter API? Was any particular library package used for extraction? Which programming tool was used? And referring back to the aforementioned point about “location,” what filter(s) were used for the extraction other than keywords?

What specific NLTK library was used for word segmentation? When returning to the original tweets of top frequently used words, what specific method(s) used to further explore discussions?

Overall, method section is weak and needs major improvements; basic details are missing and arguments/explanations move on surface for the most parts, please refer back to the aforementioned points and solve them accordingly.

Results:

Define Botometer, don’t expect every reader to have a good understanding of such measures. The very first paragraph of the result section reads very hard and is vague; please explain step-by-step on the procedure through which the tweets were classified (something missing in the method section), and explain in details how exactly this classification caused an increase in the percentage of social bots tweets.

It should have been explained in the previous sections, but what is the author(s) rationale for dividing the entire COVID period into 3 sections (time periods)? This is a major problem throughout the manuscript, as this relates to the research questions as well.

 Please consider changing the questioning tone of first paragraph of page 5 into a connecting statement to the following section.

Figure 3 can be hardly written, please consider enlarging it or changing with another more readable figure.

A lot of what is discussed toward the end of result section (section 4.3 and onwards) goes into a discussion or a finding section, please consider differentiating these from each other.

Discussion:

Regardless of its misplace, findings re discussed well and does not require any major changes. Referring back to what was mentioned about the bigger picture of social media, one suggestion here is to connect the trajectory of the current research topic within the bigger picture of social media conversations and public health as well as covid vaccination overall. Also, please consider mentioning the limitations/challenges of the current research, as well as potential future research avenues.

Author Response

Abstract and keywords:

Write a brief and summarized sentence or two about implications of the current study. It is good to see summary of important findings here in the abstract, however, it needs to be complemented by a reasonable conclusion, in a way that the reader can not only understand the rationale of the study but have some idea about what to expect in the following sections. Also, what specific method(s) are used in this research? That being explained, the abstract has rooms for improvement and needs to be enriched by the aforementioned elements.

Response: Thanks for your valuable suggestions. We have added a summary of the significance of the research in Abstract and improved Conclusions and Research methods.

Keywords look good, but could better represent the current study (e.g., Corona vaccine instead of vaccine), also consider adding some other relevant keywords such as social media analytics, Twitter, etc.)

Response: Thank you very much for your suggestion, which makes our keywords more adequate. We have added keywords (p. 1) as follows: COVID-19 Vaccine; Social Media Analytics; Twitter.

Introduction:

Introduction is somehow shallow and requires more input on the following: 1) describing the background and introducing research problem: explain in a sentence or two, the significance/implications/impacts of social media conversations on COVID vaccination, elaborating more on your very first citation (Benis et al., 2021), also briefly explain the importance of social bots in such discussion and basically in what ways they would manipulate vaccine discussions (a condensed summary of section 2.3); 2)what are specific objectives of the current study? what is the salience of your research? What are some main methods used for investigation?

Response: Thank you very much for your suggestions, which are valuable to us. Based on your suggestion, we have revised the Introduction.

  • Contents on the impact of social media and social bots on vaccination have been added, describing how social bots manipulate vaccine discussions in online discussions (pp. 1-2).
  • In the last paragraph of the Introduction, we propose the research objectives and describe the research methods. Specifically, the work examined social bots engaged in online discussions on Covid-19 vaccines on Twitter. Besides, text analysis and a python-based program were used to analyze bot-generated and human-generated tweets, thus finding out the social bot's participation in features (p. 2).

Literature review:

Literature review is written well, however, author(s) are highly encouraged to step back and assign a sub-section to social media, social media conversations, social media analytics, and other broad concepts in the first place, before starting with vaccine content and social media. A literature review on social media and social media analytics would be highly beneficial here and helps the lay reader to better understand/conceptualize the bigger picture. Another concern about the remainder of the lit review sections is that some major and highly related publications are missing and should be added accordingly (e.g., Charles-Smith et al., 2015).

Response: We highly appreciate this suggestion. We have added a new section to the literature review to discuss social media, social media analytics, and health topics(pp. 2-3):

“Social media is defined as web-based and mobile-based Internet applications allowing the creation, access, and exchange of ubiquitously accessible, user-generated contents [18]. Information interaction on social media has the characteristics of real-time and anonymity of information dissemination [19] as well as the potential to reach a large audience and disseminate information rapidly [20, 21]. These characteristics of social media make it an essential driving force for acquiring and disseminating information in different fields, e.g., in crisis management [22], politics [23], and dissemination of health information [24]. Analyzing data from social media offers new opportunities to analyze several aspects of health communication [25]. Kata found that at least 80% of Internet users seek health information online, and 16% of those seek online information about vaccination [26]. With the difficulty of communicating offline during the pandemic, more and more people are likely to use social media for information about vaccines and to express their views on social media platforms. On this background, social media can provide an additional informal source of data that can be used to identify health information not reported to medical officials or health departments and to reveal viewpoints on related topics [27].”

Methodology

Briefly explain the previous research design that was followed in the current study. What exact locations (207 among 218) author(s) pointed out to, and basically what did they mean by “location,” is this geographical coordinates used to interact with Twitter API, or country/city/etc.?

Response: Thank you for the comment. The locations we mentioned in the original text aim to explain the widespread circulation of these four vaccines (Pfizer Vaccines, Johnson & Johnson Vaccines, Moderna Vaccines, and AstraZeneca Vaccines) used as hashtags for collecting data. We do not interact with the Twitter API using these geographic coordinates. Since our presentation caused your misunderstanding, we have carefully revised it (p. 4).

How did the author(s) decide about vaccines with broadest coverage? Is this based on their own intuitions or are there specific scientific research that they can refer back to? And even more controversially, why more generally relevant hashtags such as #vaccine #covidvaccine are excluded from the search vocabulary?

Response: Thank you very much for your comment. The question you mentioned needs to be explained.

  • According to Statista, Pfizer, Moderna, Johnson & Johnson, and AstraZeneca are the four most ordered COVID-19 vaccines, widely circulated around the world.
  • We did not choose more generally relevant hashtags such as #vaccine #covidvaccine for these reasons:

For selecting hashtags, when #vaccine was used as the retrieved hashtag, the tweets' content included multiple vaccines. In other words, the public attaches the hashtag #vaccine to their tweets when discussing any vaccines, such as HPV vaccines or pet vaccines. The reason for not using #covid-vaccine and #covid vaccine is that the hashtag #covid-vaccine does not exist on Twitter. Searching with the hashtag #covid-vaccine shows tweets containing the keywords "covid" and "vaccine." Also, due to the space between "covid" and "vaccine," using #covid vaccine leads to search results with discussions related to #covid rather than covid-19 vaccines, deviating from our research direction. Using these specific hashtags is more focused on the research topic of the work, and we would like to see public discussions on more specific vaccines. More general hashtags are not used, which is our limitation. This issue will be studied in the follow-up research.

Were English tweets were extracted only, and if so, what is the author(s) rationale for excluding non-English tweets?

Response: Thanks for your comment. We exclude non-English tweets with the reasons as follows. In the dataset we obtained, the number of tweets in English accounted for 96%, and that of non-English tweets accounted for 4%. Considering that English is the main language in the Twitter platform, we exclude non-English tweets from the dataset.

How did the tweets were extracted in the first place? Was it through Twitter API? Was any particular library package used for extraction? Which programming tool was used? And referring back to the aforementioned point about “location” ,what filter(s) were used for the extraction other than keywords?

Response: Thank you so much for your insightful questions. All questions will be answered below.

  • The work used the official API data interface service provided by Twitter to collect tweets and authors. Twitter users' privacy is respected without any personal information collected and displayed.
  • We did not use any particular library package and programming tool for extraction.
  • As mentioned above, the locations (207 among 218) are to explain the widespread circulation of these four vaccines (Pfizer, Johnson & Johnson, Moderna, and AstraZeneca Vaccines) used as hashtags for collecting data. We do not interact with the Twitter API using these geographic coordinates.
  • When collecting data, we scraped all tweets under the selected hashtags. No other filters were used.

What specific NLTK library was used for word segmentation? When returning to the original tweets of top frequently used words, what specific method(s) used to further explore discussions?

Response: Thanks for your comments. We supplement these in our method description for text analysis (pp. 4-5):

  • Firstly, the high-frequency words used were summarized to understand the foci of the discussion between social bots and human users. Cipin, a professional word frequency analysis software based on Python, was applied with NLTK Library used for word segmentation. NLTK is a leading platform for building Python programs to work with human language data. It provides easy-to-use interfaces to over 50 corpora and lexical resources such as WordNet, along with a suite of text processing libraries for classification, tokenization, stemming, tagging, parsing, and semantic reasoning, wrappers for industrial-strength NLP libraries, and an active discussion forum [49].
  • Loading American National Corpus, we use NLTK chunk nltk.tokenize for word segmentation. Firstly, nltk.sent_tokenize(text) is used to complete the segmentation between sentences, and nltk.word_tokenize(sentence) used to complete word segmentation.
  • After getting the statistics of high-frequency words, select the top 5 high-frequency words. Ensure tweets containing these keywords and manually return relevant tweets with the specific analysis of what they were discussing.

Overall, method section is weak and needs major improvements; basic details are missing and arguments/explanations move on surface for the most parts, please refer back to the aforementioned points and solve them accordingly.

Response: Thanks again for your detailed and insightful advice. Your comments and suggestions are constructive and inspiring to us. We responded to your question above, involving a discussion on the limitations at the end as well (p. 13).

Results:

Define Botometer, don’t expect every reader to have a good understanding of such measures. The very first paragraph of the result section reads very hard and is vague; please explain step-by-step on the procedure through which the tweets were classified (something missing in the method section), and explain in details how exactly this classification caused an increase in the percentage of social bots tweets.

Response: Thank you for your comments. We have made modifications as follows.

  • We have added the introduction to Botometer, which is presented in detail in Section 3.2 (p. 4).
  • Here we have added the process of detecting social bots using Botometer, showing the analysis results of Botometer (p. 5).
  • We are sorry that our vague statement caused your incomprehension. The results are recognized in this section as follows: “We entered 255,277 valid tweets from 102,844 Twitter users into Botometer to get a score. Fig. 1 shows the distribution of scores returned by Botometer. Botometer detected 9,122 social bots and 93,722 human users among 102,844 Twitter users, accounting for 8.87 and 91.13% of total users, respectively. Social bots posted 28,081 and humans posted 227,196, accounting for 11% and 89% of the total tweets, respectively. (p. 5)”

It should have been explained in the previous sections, but what is the author(s) rationale for dividing the entire COVID period into 3 sections (time periods)? This is a major problem throughout the manuscript, as this relates to the research questions as well.

Response: The distribution of the number of tweets was plotted by human users and social bots. We find that this trajectory fluctuates. There are several periods where the number of tweets is on the rise. We found that the number of tweets by human users increased by more than 25% in December-January 2020, March-April 2021, and June-July 2021. In other words, they had the highest discussion heat at these three stages. So what do social bots post when human users are raving about a Covid-19 vaccine on social media? We divide these three time periods for the convenience of investigation. December 2020 to January 2021 as the first phase of discourse (Stage 1), March to April 2021 as the second phase of discourse(Stage 2), and June to July 2021 as the second phase of discourse (Stage 3).

Please consider changing the questioning tone of first paragraph of page 5 into a connecting statement to the following section.

Response: We are grateful for this suggestion. The first paragraph on page 5 is amended.“This finding proved the first question that 8.87% of users who tweeted about the COVID-19 vaccine were social bots. We analyzed tweets posted by social bots and human users to figure out what the social bots posted and how these actors influenced online Covid-19 vaccine discussions (p. 5).”

Figure 3 can be hardly written, please consider enlarging it or changing with another more readable figure.

Response: Thank you for your suggestion. We replaced the original figure with a table to present data more clearly,  (see Table 2, p. 6).

A lot of what is discussed toward the end of result section (section 4.3 and onwards) goes into a discussion or a finding section, please consider differentiating these from each other.

Response: Thanks for your comment. We have changed Results into Findings and Discussion (p. 5).

Discussion:

Regardless of its misplace, findings re discussed well and does not require any major changes. Referring back to what was mentioned about the bigger picture of social media, one suggestion here is to connect the trajectory of the current research topic within the bigger picture of social media conversations and public health as well as covid vaccination overall. Also, please consider mentioning the limitations/challenges of the current research, as well as potential future research avenues.

Response: We are grateful for this suggestion and have expanded Discussion and Conclusion with a new paragraph in Conclusion to elaborate the relationship among social bots, public health, and covid-19 vaccination (p. 13). In the last part of the work, “Limitations and Future Directions” is added to illustrate the limitations and future research directions (p.13).

The revision has gone through proofreading process again.

Reviewer 2 Report

  1. I got the impression that the authors were repeating the existing views. Social bots have been, are and will be. What's new in this article?
  2. Article has great keyword potential. I would consider adding a few keywords.
  3. Chapter 2. Literature - Isn't it better to change the name to Background or Literature review?
  4. Research questions seem rhetorical. Nowadays, bots are common on social media. It's actually obvious that bots also took part in the vaccine discussion. So the question should not be "whether the bots were in the discussion" but "how many bots were in the discussion". Please comment.
  5. Is the statement: “At present, the COVID-19 pandemic is still grim, and one of the ways to limit the epidemic is to get effective COVID-19 vaccines” is a study summary? Was that what the research was about? Is this article for vaccine promotion or social media research?
  6. Who is behind social media bots? What is the purpose of running bots and who runs them? What is it done for?
  7. Actually, I don't know what the main conclusion of this article is? What is the main research result? What is the main finding of the research?
  8. What are the practical implications of the research carried out? It is worth supplementing it. It is also worth writing about the limitations of this research and future research.

Author Response

I got the impression that the authors were repeating the existing views. Social bots have been, are and will be. What's new in this article?

Response: We are grateful for these suggestions, which prompted us to elaborate on these related issues and consider how we differ from existing research. The work enriches the existing research on social bots and online vaccine discussions and helps to comprehensively understand social bot’ characteristics and current dynamics. With the spread of the Covid-19 epidemic, vaccination against the Covid-19 has become an important means to control the epidemic. The normalization of social bots in social media may affect public perceptions of vaccine effectiveness and shape public health knowledge, complicating the climate of opinion on social media. Researchers and practitioners need to understand the engagement of social bots in social media to develop new communication strategies. Researchers and policymakers can understand how social bots are participating in Covid-19 vaccine discussions by our study. The above elaboration has been included in the Conclusion section (p. 13).

Article has great keyword potential. I would consider adding a few keywords.

Response: We greatly appreciate the detailed suggestion and have added more keywords as suggested (p. 1): COVID-19 Vaccine; Social Media Analytics; Twitter

Chapter 2. Literature - Isn't it better to change the name to Background or Literature review?

Response: We appreciate the comments here and have changed "Literature" to "Literature Review" (p. 2).

Research questions seem rhetorical. Nowadays, bots are common on social media. It's actually obvious that bots also took part in the vaccine discussion. So the question should not be "whether the bots were in the discussion" but "how many bots were in the discussion". Please comment.

Response: Thanks for your suggestions. We revised RQ1 (p. 4) as "How many social bots are participating in the COVID-19 vaccine discussion on Twitter?". Besides, a response to this question is given on page 7.

Is the statement: “At present, the COVID-19 pandemic is still grim, and one of the ways to limit the epidemic is to get effective COVID-19 vaccines” is a study summary? Was that what the research was about? Is this article for vaccine promotion or social media research?

Response: Thank you very much for your insightful comment. “At present, the COVID-19 pandemic is still grim, and one of the ways to limit the epidemic is to get effective COVID-19 vaccines” is not the summary of our study. Our research focuses on social media. Social bots are gradually becoming a common phenomenon on social media. However, the normalization of social bots in social media may affect public perceptions of vaccine effectiveness and shape public health knowledge, complicating the climate of opinion on social media. So, we need to keep abreast of the dynamics of social bots on social media. The work aims to discover the social bots existing in the discussion of the COVID-19 vaccine on Twitter, showing the characteristics of their participation in the vaccine discussion.

Who is behind social media bots? What is the purpose of running bots and who runs them? What is it done for?

Response: We are grateful for the insightful suggestions. Social bots are detected using Botometer in the work. However, while bot detection algorithms can automatically label suspicious social bots, the Twitter platform's current mechanism does not reveal the manipulators behind social bots. Therefore, although characterizing the content posted by social bots and their participation in discussions and proposing several possible explanations, we cannot infer the intentions of social bots based on their expressions. This is a limitation of our study. For this reason, a subsection is dedicated at the end of the paper to explain this limitation (p. 13).

Actually, I don't know what the main conclusion of this article is? What is the main research result? What is the main finding of the research?

Response: We greatly appreciate your insightful comments. Our main findings are as follows.

  • We found the participation of social bots in the COVID-19 vaccines discussion on Twitter and detected 28,081 (11%) tweets published by 9,122 (8.87%) social bots in the dataset. The activity level of social bots was consistent with that of human users.
  • There were three hotly debated periods for Covid-19 vaccine discussions on Twitter. These three periods are closely related to real-world events. However, while social robots and humans discuss the same topics in these three time periods, they express different meanings. Social bots are more positive in their expressions, and these findings are presented in the Conclusion sections (p. 12).

However, our study does not examine the sentiment of social bots and human users, which will be our main research direction in the future.

What are the practical implications of the research carried out? It is worth supplementing it. It is also worth writing about the limitations of this research and future research.

Response: Thank you for your constructive comments. The work shows the participation of social bots in the online discussion of the Covid-19 vaccine, which has not been studied before. Researchers and policymakers can understand the characteristics of social bots' participation in the discussion of the Covid-19 vaccine by the work, thereby developing more favorable policies. The above contents are shown in Conclusion (p. 12).

In Limitations and Future Directions (p. 13), we show the limitations and suggest future research directions. the work is one of our series of manuscripts on social robotics research. In future research, we will discuss the emotional expressions of social bots and how they affect the emotions of human users. 

Round 2

Reviewer 1 Report

The manuscript has been significantly improved when compared to the initial submission.